# Crystal structure of progeria mutant S143F lamin A/C reveals increased hydrophobicity driving nuclear deformation

Jinsook Ahn[1,3], Soyeon Jeong[1,3], So-mi Kang[2], Inseong Jo[1], Bum-Joon Park [2] & Nam-Chul Ha [1✉]

Lamins are intermediate filaments that form a 3-D meshwork in the periphery of the nuclear envelope. The recent crystal structure of a long fragment of human lamin A/C visualized the tetrameric assembly unit of the central rod domain as a polymerization intermediate. A genetic mutation of S143F caused a phenotype characterized by both progeria and muscular dystrophy. In this study, we determined the crystal structure of the lamin A/C fragment harboring the S143F mutation. The obtained structure revealed the X-shaped interaction between the tetrameric units in the crystals, potentiated by the hydrophobic interactions of the mutated Phe143 residues. Subsequent studies indicated that the X-shaped interaction between the filaments plays a crucial role in disrupting the normal lamin meshwork. Our findings suggest the assembly mechanism of the 3-D meshwork and further provide a molecular framework for understanding the aging process by nuclear deformation.

[1] Department of Agricultural Biotechnology, Center for Food and Bioconvergence, and Research Institute of Agriculture and Life Sciences, CALS, Seoul National University, Seoul 08826, Republic of Korea. [2] Department of Molecular Biology, College of Natural Science, Pusan National University, Busan 46241, Republic of Korea. [3]These authors contributed equally: Jinsook Ahn and Soyeon Jeong. ✉email: hanc210@snu.ac.kr

The nuclear lamina is mainly composed of intermediate filament (IF) lamins and forms a protein-made meshwork that maintains the nuclear shape[1–3]. Lamins consist of an alpha-helical central rod domain, an N-terminal head region, and a C-terminal long tail region. The central alpha-helical rod domain of lamins is further divided into three subdomains (i.e., coil 1a, coil 1b, and coil 2) that are interrupted by linkers L1 and L12. Lamins belong to the type V IF family, distinguished from the cytosolic types of IF proteins such as vimentin and keratin[4,5]. The C-terminal tail region contains a nuclear localization signal and a globular Ig-like domain, which are unique features of lamins compared to cytosolic IF proteins[6,7]. Another feature of vertebrate lamins is the additional six-heptad repeats in the middle of the coil 1b regions[8–10].

A recent in situ cryo-ET structure of lamin filaments revealed a 3.5-nm-thick filamentous structure that is regularly decorated with Ig-like domains. The lamin filaments were significantly thinner than the cylindrical-shaped cytosolic types of IF proteins with a 10-nm diameter[11,12]. The basic building units of lamin filaments are coiled-coil dimers formed by parallel coiled-coil interactions between the central rod domains. The two basic building units are further assembled in an antiparallel manner by overlapping the coil 1b regions, called the "A11 tetramer", as a polymerization intermediate. The crystal structure of a long N-terminal fragment containing two-thirds of the central rod domain of human lamin A/C visualized high-resolution features of the A11 tetramer, essentially formed by the A11 interaction[9,13–15]. To longitudinally join the A11 tetramers, the other interaction (i.e., the so-called A22 interaction), is necessary. The A22 interaction describes the antiparallel interaction between the coil 2 regions. According to the previously proposed assembly model[9], the alternative and mutual synergistic application of the A11 and A22 interactions to the basic building dimeric units created the 3.5-nm-thick filamentous structure of lamin A/C. In the process of the A22 interaction, the N- and C-terminal of lamin A/C overlap in parallel, and the lengths of the repeating rod domain were proposed to be 40–51 nm[16]. The biochemical evidence using the cross-linking mass analysis proposed a compression mechanism to explain the intrinsic elasticity of the lamin filaments[16]. The model based on the in situ cryo-ET structure supports 40-nm repeating units with 14-nm overlapping when the Ig-like domains were used as references[9,11]. High-resolution structures may be required to understand filament formation at the molecular level.

The structural organization of intermediate filaments is essential in understanding IF-related human diseases. However, only a few mechanisms have been proposed to structurally connect genetic mutations of IF proteins and human diseases. Structural analysis of the K1/K10 heterotetramers demonstrated that a single mutation of type I keratins alters the electrostatic surface potential of the K1/K10 heterotetramers, which causes human diseases. Several genetic mutations of lamin A/C result in defects in the nuclear envelope, leading to progeroid syndrome or muscular dystrophies[17–20]. The S143F mutation at *LMNA* exhibited mixed phenotypes of muscular dystrophy and progeroid syndrome with a deformed nuclear envelope structure, as observed in HGPS cells[21–23]. Hutchinson Gilford progeria syndrome (HGPS) is caused by a dominant mutation at the splicing site in *LMNA*, resulting in the expression of progerin, whose C-terminal tail is altered from normal lamin A. Progerin remains farnesylated at the altered C-terminal end, unlike the mature form of wild-type lamin A/C[24]. Many mechanisms have been proposed to structurally connect genetic mutations and nuclear deformation. The remaining farnesyl group was proposed to be the main reason for the nuclear deformation induced by progerin[24–26]. However, our research group has suggested that an aberrant additional interaction between the Ig-like domain and the distinct C-terminal region of progerin is critical for nuclear deformation[27–29].

This study determined the crystal structure of the long fragment of lamin A/C harboring the S143F mutation. Based on the crystal structure and subsequent biochemical studies, we propose a molecular mechanism accounting for nuclear deformation, the hallmark of progeroid syndromes, at the molecular level.

## Results

**The overall structure of the lamin S143F mutant protein.** We overproduced the S143F mutant lamin A/C fragment (residues 1–300, referred to as the lamin 300 fragments) in an *E. coli* expression system. The mutant lamin protein behaved as a dimer similar to the wild-type lamin 300 fragment, as observed in a previous report (Supplementary Fig. 1)[9]. Crystals of the mutant lamin belonged to the $P4_122$ space group. We determined the crystal structure at 3.7 Å resolution by molecular replacement using half pieces of the wild-type lamin 300 fragment structure (PDB code: 6JLB)[9]. The asymmetric unit contained one coiled-coil dimer, where the subdomains coil 1a, L1, coil 1b, and L12 regions were well ordered (residues 29–229 in one chain and residues 31–229 in the other chain) (Fig. 1 and Supplementary Fig. 2).

The coiled-coil dimer of the S143F mutant lamin showed long alpha-helix bending at linker L1 between coil 1a and coil 1b (Fig. 1). Superposition of the S143F mutant and A:B or C:D coiled-coil dimers of the wild-type lamin structures have a root-mean-square deviation (RMSD) of 2.3 or 4.2 Å, respectively. A comparison to the wild-type lamin coiled-coil dimer structures showed that the bending directions of coil 1a are different with kinks at linker L1, reflecting conformational fluctuations of coil 1a and linker L1 with respect to coil 1b (Supplementary Fig. 3). The mutated Phe143 is at the "c" position of the heptad repeat in coil 1b, orientating to the outside of the coiled-coil structure; thus, the mutation did not result in a substantial change in the coiled-coil structure of coil 1b (Supplementary Figs. 3, 4a).

**The S143F mutation does not change the interactions to make the linear filament.** The crystallographic twofold symmetry built the A11 tetramer, showing the antiparallel arrangement of the two dimers by overlapping the coil 1b region (Fig. 2). Furthermore, we confirmed that the S143F mutant lamin does not interfere with the A11 interaction in forming the linear filament

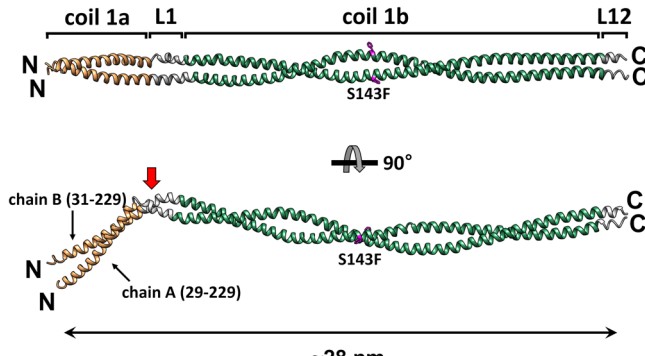

**Fig. 1 Crystal structure of the S143F mutant lamin 300 fragments (residues 1–300).** The asymmetric unit structure of the S143F mutant lamin fragment consists of two chains in a parallel coiled-coil arrangement. The ordered residues and the length of the coiled-coil dimer are indicated. Coil 1a is in yellow, linkers L1 and L12 are in gray, and coil 1b is in green. The kinks are in linker L1, as indicated by the red arrow. The Phe143 residues are colored magenta in stick representation.

since the A11 tetrameric structures between S143F and wild-type lamin A/C were similar overall, with an RMSD of 3.9 Å (Fig. 2 and Supplementary Fig. 4b).

The A11 tetramers further assemble to elongate the filament via the A22 interaction, representing the antiparallel arrangement between the entire coil 2 regions, including the coil 1a part. The direct and robust interaction between the insertion of the C-terminal region of coil 2 into the coil 1a region of the lamin 300 fragment was the key binding for the A22 interaction (Fig. 3a)[9]. To determine the mutational effect of S143F in terms of the A22 interaction, we probed the A22 interaction using the C-terminal fragment of coil 2 (residues 250–400; called lamin 2 C fragment) with the wild-type or S143F mutant lamin 300 fragments. The obtained results demonstrated that the S143F

mutation did not affect the A22 interaction under two buffer conditions; the lamin 300 fragment was strongly bound to the C-terminal fragment of coil 2 at a relatively low NaCl concentration of 50 mM and showed weak binding to the C-terminal fragment of coil 2 at a relatively high NaCl concentration of 150 mM (Fig. 3b). This finding was not surprising because Ser143 is not involved in the A22 binding interaction in the lamin 300 fragments (Fig. 3).

**Phe143-mediated X-shaped interaction between the A11 tetramers.** We examined two A11 tetramers that directly interacted in the crystal because significant protein–protein interactions were observed under highly condensed conditions in the crystal (Fig. 4). Of note, the crystal structure of the S143F mutant lamin showed an X-shaped interaction between the two tetrameric units around Ala146 at the central residue of the tetramer (Supplementary Fig. 4b). This X-shaped interaction between the A11 tetramers was not observed in the wild-type structure of the same lamin 300 fragment (Ahn, Jo et al. 2019)[9]. Of note, we found that the mutated Phe143 residues were clustered on the surface of the X-shaped structure with the surface-exposed leucine residues Leu140, Leu141, Leu148, and Leu152 (Fig. 4). Because the leucine residues were sandwiched between the A11 tetramer, the hydrophobic residues were four times more populated than the monomeric form of lamin.

We calculated the surface hydrophobicity of the X-shaped contacting region (residues 119–171) between the A11 tetramers using the Protein-Sol server[30]. The S143F mutant protein exhibited maximum hydrophobicity at the contact region around the Phe143 residue (3.94; corresponding to the hydrophobic association between heavy and light chains in the Fab domain). However, the corresponding contact region of the wild-type protein was less hydrophobic than the peripheral region (2.66 at the peripheral region; Fig. 5a, b). Furthermore, the obtained structure suggested that Phe143 exceeded the threshold to make the X-shaped interaction by adding hydrophobicity to the binding

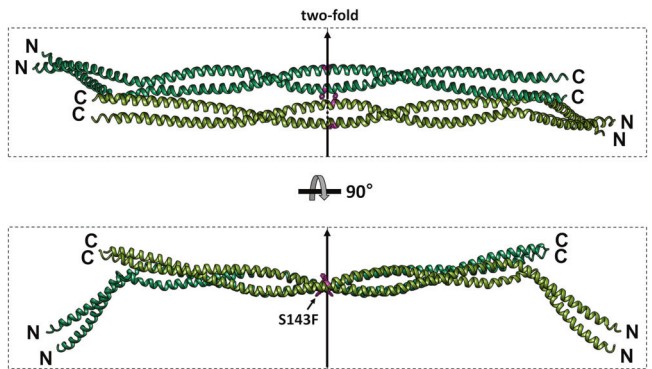

**Fig. 2 The tetrameric unit of the S143F mutant lamin 300 fragments.** The A11 tetramer structure was generated by applying crystallographic twofold symmetry to the asymmetric unit. The top and side views of the tetramer, consisting of two antiparallel coiled-coil dimers, are colored separately. The twofold symmetry axis is represented by a black arrow. The Phe143 residues are colored magenta in stick representation.

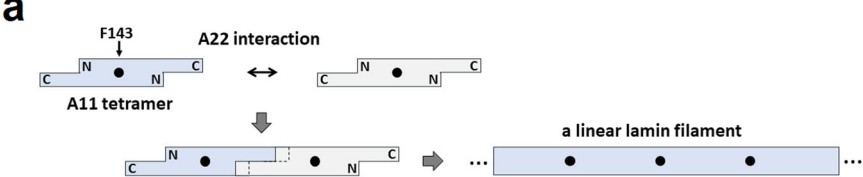

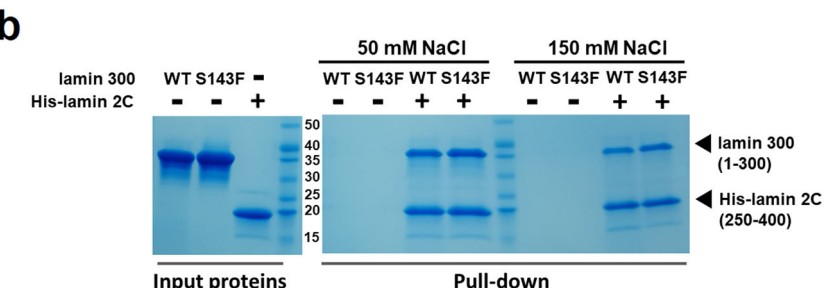

**Fig. 3 Effects of the S143F mutation on the A11 and A22 interactions. a** A schematic drawing representing linear lamin filament formation by the A11 and A22 interactions. The diagram with two wings represents the A11 tetramers of the full-length lamin, where the black dots indicate Phe143 in the S143F mutant lamin A/C. The successive longitudinal overlap between the tetramers by the A22 interactions forms a long linear filament, resulting in repetitive Phe143 sites with 40-nm intervals. **b** SDS-polyacrylamide gel showing the direct binding between lamin 300 and lamin 2 C fragment. The amounts of the protein samples were measured by SDS–PAGE (input proteins). The binding between the lamin 300 fragment (WT and S143F) and the hexaHis-tagged lamin fragment (His-lamin 2 C fragment; residues 250–400) was analyzed by pull-down experiments using Ni-NTA resin. The tagless lamin 300 fragments were incubated on the empty resin (−) or hexaHis-tagged protein-bound resin. Then, the resins were pre-equilibrated and washed with 20 mM Tris-HCl (pH 8.0) buffer containing 20 mM imidazole and 50 mM or 150 mM NaCl for pull-down, followed by SDS–PAGE analysis.

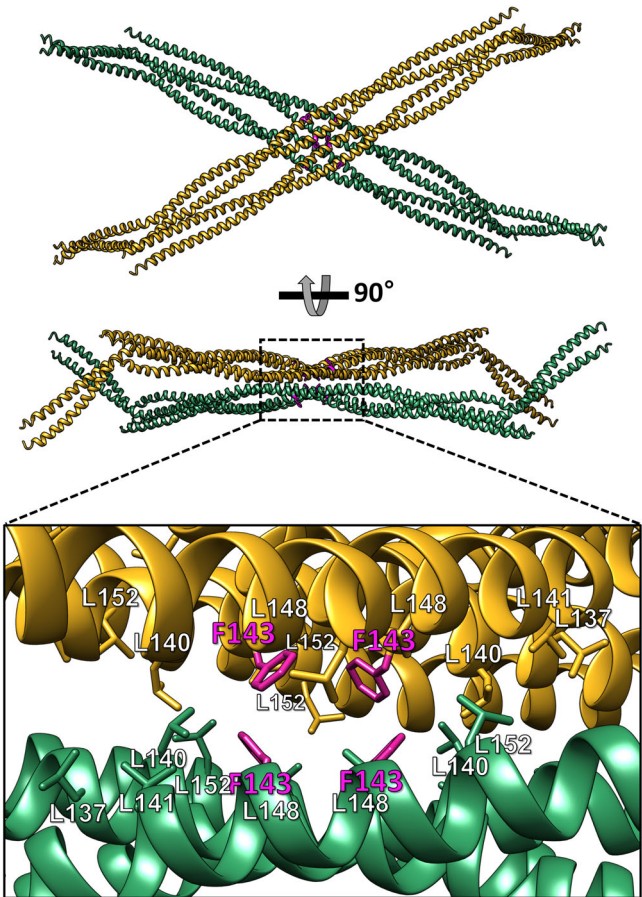

**Fig. 4 The X-shaped interaction between the two A11 tetramers of the S143F mutant lamin fragment in the crystals.** The top and side views of the X-shaped interaction between the two A11 tetramers were generated by crystal packing interactions. The Phe143 residues are presented in the magenta stick representations. The hydrophobic residues on the contact region for the X-shaped interaction between the two tetramers are shown in stick representations in the bottom panel.

interface in the center of the existing hydrophobic Leu residues on that tetramer (Fig. 4). Moreover, it was determined that the hydrophobic surface area of the S143F mutant structure was increased by 1.5% compared to that of the wild-type structure by calculating the solvent-accessible surface area (SASA; Fig. 5c).

We experimentally compared the surface hydrophobicities of the wild-type and S143F mutant lamin 300 fragments using hydrophobic interaction chromatography. The obtained result showed a higher hydrophobicity on the surface of the S143F mutant fragment than that of the wild-type fragment (Fig. 5d). Thus, it is reasonable that the mutated Phe143 residues played a critical role in triggering the X-shaped interaction between the A11 tetramers.

**Synergistic aggregation of the S149C mutation near Phe143 in the cell**. Next, we investigated the effects of the S143F mutations based on the full-length lamin protein in the filament forms in HT1080 cells. HT1080 cells have been widely used in the observation of nuclear morphology because the nuclei of HT1080 cells were relatively large enough to observe the nuclear shapes. The ectopic expression of the S143F mutant lamin A/C in HT1080 cells created the irregular aggregation of lamin A/C near the nuclear envelopes, as observed in previous reports (Fig. 6)[22,31]. We noted Ser149 near Phe143 on the surface of the X-shaped structure (Supplementary Fig. 5). Although the S143F mutant

structure may have a limitation in resolution in accurately placing the side chains, the electron density map showed that the Ser149 residues are close enough to the formation of a disulfide bridge between adjacent chains (Supplementary Fig. 2). We changed Ser149 to cysteine to mimic the disulfide-mediated X-shaped interaction because Ser143 and Ser149 are on the same interface for the X-shaped interaction (Supplementary Fig. 5). When lamin A/C harboring a mutation of S143F or S149C was transfected into the cells, the cells showed nuclear deformation with deposition in the cytosol, unlike the wild-type lamin, which indicated that S149C mutant lamin forms a higher oligomeric form in cells (Fig. 6). Western blotting analysis of the ectopically expressed lamin protein in the cells showed that the S149C and S143F/S149C mutations formed disulfide bond-linked oligomers. This result implied that Cys149 is within a distance to make a disulfide bond in the lamin aggregates, which is compatible with the X-shaped interaction between the lamins in our S143F structure (Supplementary Figs. 5, 6). Thus, our results suggest that the surface containing the Ser143 and Ser149 residues may be the platform for aggregating full-length lamin A/C in the cell when it is amplified between adjacent lamin filaments.

## Discussion

The currently prevailing model for HGPS has suggested that nuclear deformation results from the unprocessed farnesyl group of the altered C-terminal end of progerin[32,33]. However, the S143F mutant lamin A/C also produced a deformed nuclear envelope structure even though the mutant lamin A/C does not contain the farnesyl group. Therefore, additional or other molecular mechanisms may exist in the nuclear deformation process caused by the progeroid syndrome. This study confirmed that the S143F mutation did not affect the A11 and A22 interactions, which are the main forces for the linear filaments. Furthermore, these results indicate that the S143F mutation did not interfere with the 3.5-nm-thick linear filaments in the cell.

How could the S143F mutation cause local aggregation of lamin filaments around the nuclear cell membrane with nuclear deformation, as observed in progeroid syndrome cells? To answer this question, we need to expand our understanding of the lamin structure from linear filaments to the 3-D meshwork structure. Our crystal structure of the S143F mutant lamin showed new molecular contacts around the substituted Phe143 residues, strengthening the X-shaped interaction between the A11 tetramers or the linear lamin filaments. Of note, the X-shaped interaction between the filaments destroyed the 3-D meshwork structure of the nuclear lamin filament network. In the crystal structure of keratin (K1/K10 and K5/K14), a similar X-shaped interaction between heterodimers mediated by the disulfide bond was observed, which was important in the formation of the normal keratin structure, unlike the S143F mutation in lamin. Interestingly, disulfide-bonded K14 was concentrated around the nucleus[34,35]. Removal of the X-shaped interaction in keratin K14 by mutation of C367A affected the size and shape of keratinocyte nuclei[35]. These results indicated that the X-shaped structure mediated by the disulfide bond in keratin was essential in maintaining the three-dimensional shapes of the nucleus.

We believe that the aberrant X-shaped interaction between the lamin filaments is critical in understanding the S143F mutational effect. Phe143-containing hydrophobic surfaces present every A11 tetramer along the filament axis in the molecular model for the linear lamin filament (Figs. 3a, 7)[9]. This regular repeatability of the hydrophobic surfaces would result in positive cooperativity in the X-shaped interaction between the filaments (Fig. 7). If two linear filaments have the X-shaped interaction on the surface patch, the interaction would be transient because only one

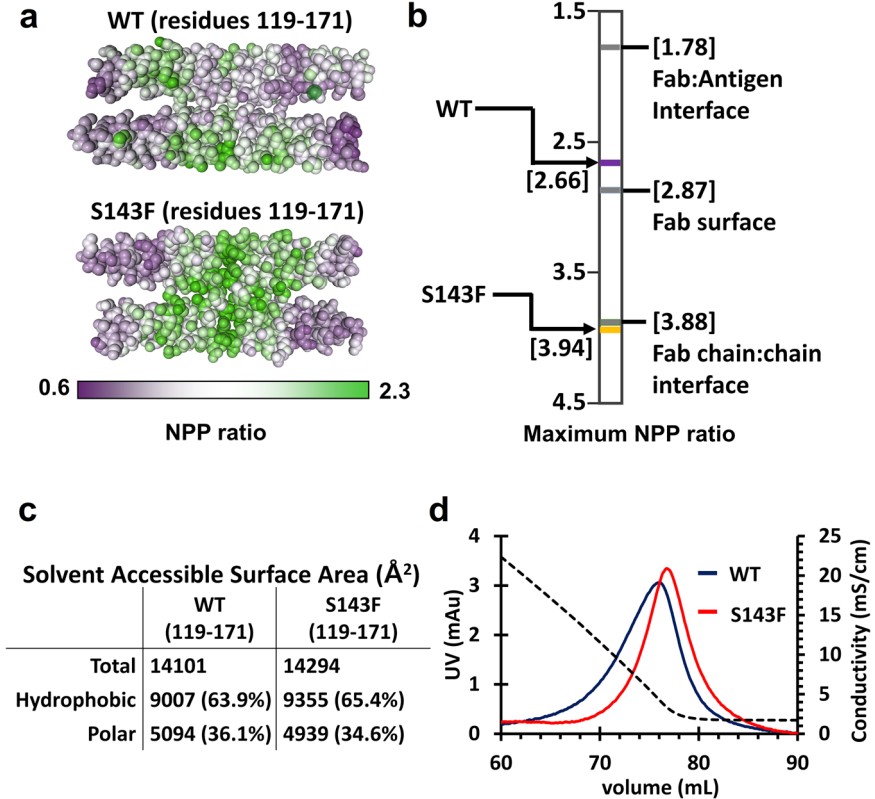

**Fig. 5 Calculation of the surface hydrophobicity around the X-shaped contacting region (residues 119–171) of the wild-type and S143F mutant lamin A/C. a** Residues 119–171 were used to calculate the surface hydrophobicities of the wild-type and S143F mutant lamins. The values of the local nonpolar to a polar ratio (NPP) are represented on the surfaces of the lamin proteins, color-coded from low NPP ratio (purple) to high NPP ratio (green). **b** The maximum NPP values of the X-shaped contacting region on the lamin were calculated using the Protein-Sol server[30]. The standard maximum NPP values are displayed on the bar. The lamin S143F mutant has a higher surface hydrophobicity than the surfaces between the heavy and light chains of antibody Fab (Fab chain:chain interface). **c** Comparison of the solvent-accessible surface areas in the X-shaped contacting region of the wild-type and S143F mutant lamins. **d** Profiles of hydrophobic interaction chromatography of the wild-type and S143F mutant lamin 300 fragments. The UV curves of wild-type and S143F are represented as blue and red solid lines, respectively. The conductivity curve resulting from the decreasing gradient of the ammonium sulfate concentration is shown as the black dotted line.

contact site exists between the two filaments (Fig. 7i, ii). However, if four filaments are involved, every filament would make two contact sites, mutually enhancing the X-shaped interaction between the filaments (Fig. 7iii, iv). Thus, once the X-shaped interaction occurs at a region of the nuclear lamina, it would be generated gradually on the linear lamin filaments in a positive-feedback manner (Fig. 7i–v). The localized lamin deposition produced by the regularly repeated X-shaped interactions between the filaments would provide uneven rigidity on the nuclear lamina. The uneven rigidity may be the molecular reason for the blebbing or local aggregates of the lamin filaments in nuclear envelopes (Fig. 7vi).

In the type I keratin K1/K10 tetramer, the genetic mutation S233L increased hydrophobicity on the K1/K10 tetramer surface[15,34,36]. This aberrant extra hydrophobicity induced higher-order aggregation, resulting in epidermolytic palmoplantar keratoderma[15,36]. This mechanism shows that a single-residue change in IF proteins may result in a significant effect on filament formation in the pathogenesis of human diseases. Thus, this observation at the keratin mutation is analogous to the lamin S143F mutation.

We propose that aberrant interfilament interactions are a common molecular cause between the S143F mutant lamin and progerin in progeroid syndrome. Progerin exhibited an aberrant interaction between the altered C-terminal end and the Ig-like domain of progerin or wild-type lamin A/C, both of which

existed regularly and repeatedly in the filaments[27–29]. Because the C-terminal alteration at progerin is unlikely to interfere with linear filament formation, the aberrant interaction of progerin would cause the X-shaped interaction between the filaments. Because both cases are formed between the filaments containing the regular and repeated binding sites, the interactions would be amplified in a positive-feedback manner, resulting in the local aggregation of the lamin filaments, which is the hallmark progeroid syndrome.

In this study, we visualized the aberrant X-shaped interaction by S143F, which abolished the normal 3-D meshwork between the lamin filaments. Our findings suggest that abnormal 3-D meshwork formation may be responsible for the local aggregation of linear lamin filaments, leading to the deformation of the nuclear shapes. We further presented "strengthened molecular interactions between the linear lamin filaments" as a common mechanism to explain aging-related nuclear deformation.

## Methods

**Plasmid construction**. We used plasmids expressing wild-type lamin fragments (residues 1–300 and residues 250–400), as previously reported[9]. We performed PCR-based site-directed mutagenesis on the wild-type lamin fragment (residues 1–300) to overexpress the S143F mutant lamin fragment (residues 1–300). The resulting DNA was inserted into the pProEx-HTa vector (Thermo Fisher Scientific, MA, USA). The resulting plasmid was transformed into the *Escherichia coli* strain BL21 (DE3; Novagen, USA) or B834 (DE3; Novagen, USA) to obtain the selenomethionyl-labeled protein for crystallization.

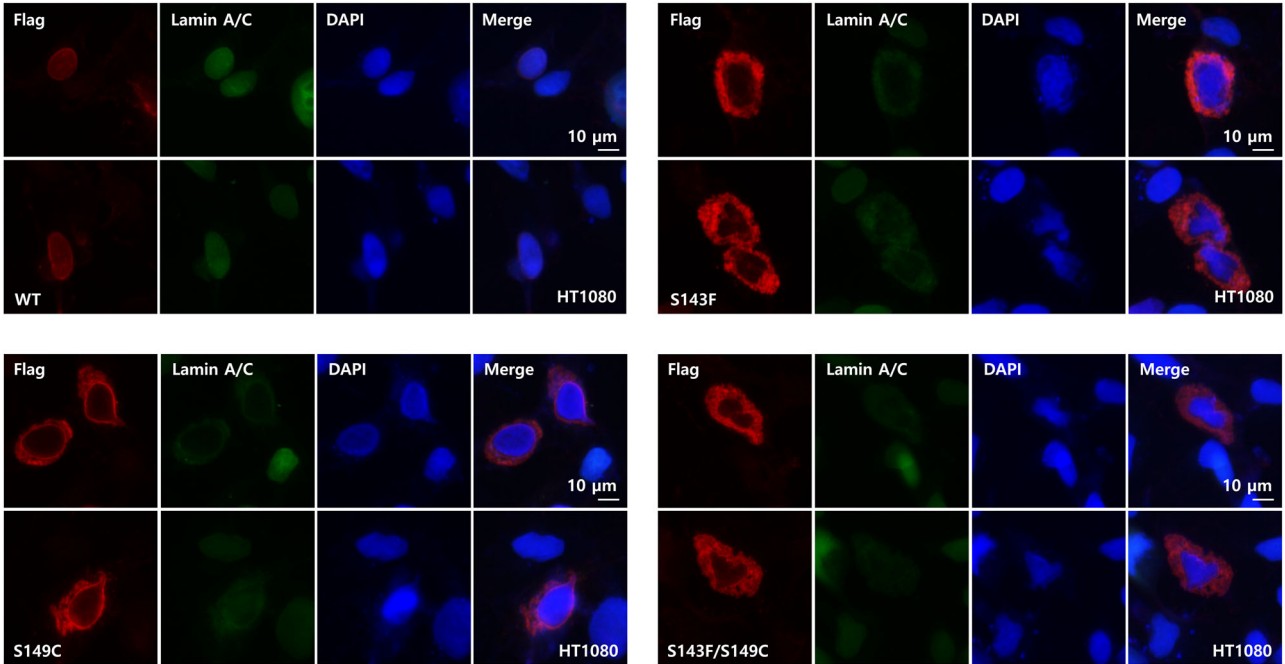

**Fig. 6 Nuclear shapes and distribution of wild-type and S143F lamin A in HT1080 cells.** Immunofluorescence assays were performed to visualize the nuclear morphology after the transfection of the vector overexpressing wild-type or S143F Flag-tagged lamin A into HT1080 cells. For visualization of the nuclear membrane, cells were stained with an anti-Flag antibody for Flag-tagged lamin A (red), anti-lamin A/C antibody for total lamin A/C (green), and DAPI for DNA (blue). The merged images of lamin A/C and DNA are displayed on the right (merge). Scale bar: 10 μm.

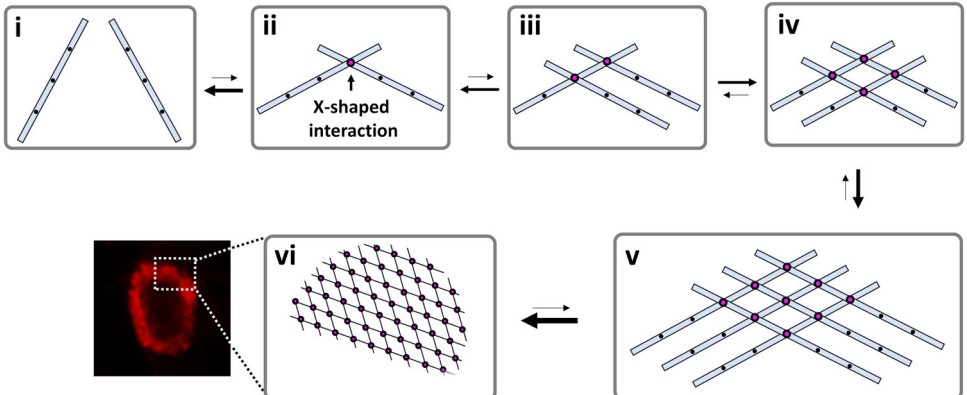

**Fig. 7 The proposed mechanism for the aggregation of lamin filaments by the S143F mutation.** A hypothetical mechanism is proposed to explain the mutual amplification of Phe143-triggered hydrophobic interactions. The two linear filaments form the reversible X-shaped interaction on the Phe143 sites (magenta circle) at low affinity (i, ii). Another linear filament is joined to the existing filaments at a similar affinity on the Phe143 site near the preoccupying site, resulting in similar binding sites for recruiting the other lamin filaments (iii). Finally, the fourth filament is bound to the two Phe143 sites simultaneously, resulting in a mutual binding platform at a higher binding affinity (iv). The equilibrium is increasingly shifted to the association as more filaments are recruited. All linear filaments make contacts on more Phe143 sites (v), resulting in local aggregation on the nuclear membrane (vi).

**Purification of the recombinant proteins**. Transformed *E. coli* cells were cultured in 4 L of Terrific broth or M9 medium supplemented with L-(+)-selenomethionine at 37 °C. Protein expression was induced by 0.5 mM IPTG at 30 °C. After the cells were harvested by centrifugation, they were resuspended in lysis buffer containing 20 mM Tris-HCl (pH 8.0), 150 mM NaCl, and 2 mM 2-mercaptoethanol. The cells were disrupted using a continuous-type French press (Constant Systems Limited, Daventry, UK) at 23 kpsi pressure, and the cell debris was removed by centrifugation at $19,000 \times g$ for 30 min at 4 °C, after which the supernatant was loaded onto cobalt-Talon affinity agarose resin (Qiagen, The Netherlands) in lysis buffer. The target protein was eluted with lysis buffer supplemented with 250 mM imidazole. The eluate was treated with TEV protease to cleave the hexaHis-tag and was then loaded onto a HiTrap Q column (GE Healthcare, USA). A linear gradient of increasing NaCl concentration was applied to the HiTrap Q column. The fractions that contained the protein were applied onto a size exclusion chromatography column (HiLoad Superdex 200 26/600 column; GE Healthcare, California, USA) pre-equilibrated with lysis buffer.

**Crystallization, structure determination, and analysis**. The S143F mutant lamin 300 fragment (8 mg/ml) was crystallized in a precipitation solution containing 0.2 M Tris-HCl (pH 8.0), 0.1 M sodium citrate, and 18% PEG 400 using the hanging-drop vapor diffusion method at 16 °C. Next, the crystals of the lamin fragment were briefly dipped in a cryoprotectant solution containing 20% galactose and flash-frozen in a nitrogen stream at −173 °C. The diffraction datasets were collected using an Eiger 9 M detector at Beamline 5 C at PLS (Pohang, Republic of Korea) and were processed with the HKL2000 package[37]. The structure of the S143F mutant lamin fragment was determined using the molecular replacement method with MOLREP in the CCP4 package[38]. The structure of the wild-type lamin 300 fragment (PDB code: 6JLB) was split into two parts (residues 67–145; 146–240) and was used as the search model for molecular replacement. The crystals belonged to space group $P4_122$ with unit-cell dimensions of $a = 123.8$ Å and $c = 458.5$ Å (Table 1). The programs COOT and PHENIX were used for model building and refinement[39,40]. Statistical information regarding data collection and processing is presented in Table 1. Structural analysis was performed using COOT,

**Table 1 X-ray diffraction and refinement statistics.**

| Data collection | Lamin S143F (residues 1–300) |
|---|---|
| Beamline | Pohang Accelerator Laboratory Beamline 5 C |
| Wavelength (Å) | 0.9794 |
| Space group | $I4_122$ |
| Cell dimensions | |
| $a, b, c$ (Å) | 123.8, 123.8, 458.5 |
| $\alpha, \beta, \gamma$ (°) | 90, 90, 90 |
| Resolution (Å) | 48.1–3.7 (3.76–3.70)* |
| No. of reflections | 18666 |
| $R_{pim}$** | 0.02 (0.19)* |
| $I/\sigma I$ | 19.2 (2.4)* |
| CC(1/2) in outer shell (%) | 33.9 |
| Completeness (%) | 94.7 (80.5)* |
| Redundancy | 9.2 (2.6)* |
| **Refinement** | |
| Resolution (Å) | 48.1–3.7 (3.76–3.70)* |
| No. reflections | 16492 |
| $R_{work}/R_{free}$*** | 0.287/0.304 |
| Total no. of atoms | 3271 |
| No. ligands | 0 |
| No. water molecules | 0 |
| Wilson B-factor (Å²) | 11.1 |
| RMS deviations | |
| Bond lengths (Å) | 0.003 |
| Bond angles (°) | 0.563 |
| Ramachandran plot | |
| Favored (%) | 99.75 |
| Allowed (%) | 0.25 |
| Outliers (%) | 0.0 |
| PDB ID | 7D9N |

\* Values in parentheses are for the highest resolution shell.
\*\*$R_{p.i.m.} = \Sigma_{hkl} (1/(n-1))^{1/2} \Sigma_i | I_i(hkl) - [I(hkl)]|/\Sigma_{hkl}\Sigma_i I_i(hkl)$. $R_{pim}$ is the precision-indicating (multiplicity-weighted) $R_{merge}$.
\*\*\*$R_{free}$ calculated for a random set of 9.9% of reflections not used in the refinement.

PyMOL, and UCSF Chimera. The solvent-accessible surface area of the lamin wild-type and S143F mutant structures was calculated by the GETAREA server.

**Pull-down assays**. A pull-down assay was conducted using a hexaHis-tagged lamin fragment (residues 250–400) immobilized on Ni-NTA resin as bait. The hexaHis-tagged cleaved lamin proteins as prey were incubated with His-tagged lamin immobilized resin pre-equilibrated in 20 mM Tris-HCl (pH 8.0) buffer containing 150 mM NaCl (or 50 mM NaCl) at room temperature for 30 min. After washing with buffer containing 20 mM Tris-HCl (pH 8.0), 150 mM NaCl, and 20 mM imidazole, the fractions were analyzed using SDS–PAGE.

**Immunofluorescence staining**. A human fibrosarcoma cell line (HT1080) obtained from ATCC was maintained at 37 °C in liquid Dulbecco's modified Eagle's medium (DMEM) containing 10% (v/v) FBS and 1% (v/v) antibiotics. HT1080 cells were seeded on glass coverslips and transfected with vectors expressing wild-type and mutant full-length lamin A using jetPEI (Polyplus Transfection). Cells were rinsed briefly in PBS 24 h after transfection. After fixing with 99% methanol (stored at −20 °C for at least 2 h before use) for 15 min at −20 °C, the cells were permeabilized with 0.1% (v/v) Triton X-100 for 5 min and incubated with blocking buffer [5% normal goat serum (31873; Invitrogen) in PBS] for 1 h at room temperature. After washing briefly with PBS, the cells were incubated with an anti-lamin A/C (sc-376248; Santa Cruz Biotechnology) primary antibody (1:200; diluted in blocking buffer) in blocking buffer overnight at 4 °C, followed by a secondary antibody (anti-mouse Ab-FITC; 1:400; diluted in blocking buffer) for 7 h at room temperature in the dark. The cell nuclei were stained with DAPI for 10 min. After quickly washing three times in PBS, coverslips were applied with an antifade mounting medium. Immunofluorescence signals were detected by fluorescence microscopy (Logos).

**Western blot analysis**. After transfection with wild-type lamin A and mutant (S143F, S149C, or S143F/S149C) expression vectors into HT1080 cells, proteins were extracted from cells using radioimmunoprecipitation assay (RIPA) buffer (50 mM Tris-Cl pH 7.5, 150 mM NaCl, 1% NP-40, 0.1% SDS, and 10% sodium deoxycholate). Samples (with or without heating inactivation) were separated by sodium dodecyl sulfate–polyacrylamide gel electrophoresis (SDS–PAGE) and transferred to polyvinylidene difluoride (PVDF) membranes. Blotted membranes were blocked with 3% skimmed milk in TBS-T buffer (20 mM Tris pH 7.6, 150 mM NaCl, and 0.05% Tween 20) for 1 h followed by incubation with anti-flag antibody (1:3000; F3165; Sigma Aldrich, USA). Horseradish peroxidase (HRP)-conjugated goat anti-mouse IgG antibody (Pierce, Thermo Fisher Scientific, Inc., USA) was used as a secondary antibody. Peroxidase activity was detected by chemiluminescence using an ECL kit (Intron, Korea) following the manufacturer's instructions.

**Hydrophobic interaction chromatography**. To compare hydrophobicity, we performed hydrophobic interaction chromatography (HIC) using a Hitrap Phenyl Sepharose HP (5 mL) column (Cytiva, UK). The wild-type and S143F mutant His-tagged lamin 300 fragments were prepared in 20 mM Tris-HCl (pH 8.0) buffer supplemented with 600 mM ammonium sulfate for binding to the column. The HIC was conducted by decreasing the ammonium sulfate gradient from 600 mM to 0 mM in 20 mM Tris-HCl (pH 8.0) buffer.

**SEC-MALS**. The molecular size of lamin proteins was determined with analytical size exclusion chromatography coupled with multiangle light scattering (SEC-MALS). Samples (wild-type and S143F mutant of the lamin 300 fragment; 2 mg/mL) were injected into a Superdex 200 Increase 10/300 GL column (GE Healthcare), pre-equilibrated with a buffer containing 20 mM Tris-HCl (pH 7.5) and 150 mM NaCl. SEC-MALS data were calculated by the ASTRA 6 software (WYATT, USA).

**Reporting summary**. Further information on research design is available in the Nature Research Reporting Summary linked to this article.

## Data availability

The crystal structure of Lamin S143F (residues 1–300) is deposited at the Protein Data Bank (PDB) with PDB ID 7D9N. PDB structure coordinates are provided in Supplementary Data 1. X-ray diffraction data is provided as an MTZ file in Supplementary Data 2.

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

## Acknowledgements
This work was supported by a grant from the National Research Foundation of Korea (2019R1A2C208513512). This research was also supported by the Basic Research Program through the National Research Foundation of Korea (NRF), funded by the MSIT (2020R1A4A101932211). This research was supported by the Basic Science Research Program through the National Research Foundation of Korea (NRF) funded by the Ministry of Education (2021R1I1A1A01049976). We would like to thank the Pohang Accelerator Laboratory Beamline 5 C (Pohang, Republic of Korea) for help with X-ray diffraction experiments and Dr. Eunha Hwang (Korea Basic Science Institute, Republic Korea) for help with the SEC-MALS experiments.

## Author contributions
J.A. and S.-m.K. performed the experiments. J.A., S.J., I.J., B.-J.K., and N.-C.H. conceived the experimental designs and interpretation of data and revised the manuscript.

## Competing interests
The authors declare no competing interests.
