## [Peer Review File · Communications Biology]

Reviewers' comments:

Reviewer #1 (Remarks to the Author):

The authors report a crystal structure of a lamin A/C mutant, S143F, and show that the enhanced hydrophobicity from the Phe residue causes aberrant inter-filament interactions between lamin IFs leading to consequences in nuclear lamina architecture in the cell. Conceptually, this is an important study as the number of atomic resolution structures of IFs harboring physiologic mutations is very limited. Thus, molecular insights into Hutchinson Gilford progeria syndrome or muscular dystrophy-related mutations is clinically significant.

I think where this manuscript struggles is the depth of the analysis and the depth of the scientific discussion, which is less than the prior group's 2019 Nature Communications publication. I recommend the authors think about and address the following issues:

1. I think this manuscript erroneously uses the term "Crossing-over." When I think of "crossing-over" in the scientific sense, I think of homologous recombination of DNA, where there is intercalation of DNA strands and exchange of genetic material. There is no evidence in the figures or data provided here that lamins truly "cross-over" or intercalate their helices or exchange any type of macromolecular material. What there is evidence of is increased hydrophobicity at the molecular surface due to a missense mutation, which then drives aberrant higher-order multimerization of the lamin compared to wild-type lamin. This aberrant multimerization leads to altered nuclear lamina and deformed nuclei.

In thinking about this issue in the larger context of IF biology, and using the authors' Figure 7 diagrams as a baseline, what is really described here is an "X-shaped lamin assembly process" or "X-shaped lamin multimerization" or "X-shaped lamin oligomerization." This X-shaped phenomenon has been seen in IFs before – the Coulombe group (K5/K14) and the Bunick group (K1/K10) demonstrated with X-ray crystal structures of 2b domains of heteromeric keratins that the type I keratins contain cysteines that lead to X-shaped oligomerization. The Coulombe group further showed this altered X-shaped K5/K14 has consequences for nuclear shape.

I think the authors miss a big opportunity here to discuss more deeply the X-shaped structures in IF biology, their significance to the nucleus, and how their lamin structure is similar and dissimilar to the keratins (for example, the keratin and lamin data both have affect on the nucleus, which speaks to the importance of IFs in maintaining not just cell integrity, but nuclear integrity; they are different in that the keratin X-shapes come from disulfide bond formation, whereas the lamin is increased hydrophobic surface-exposed area from the aromatic mutant residue). Furthermore, the discussion of the disulfide data in the keratin work helps justify the authors decision to introduce S149C into their structural analysis.

I encourage the authors to think about this, and re-phrase "crossing-over" to something that genuinely reflects the work to date in the IF field. This X-shaped aggregation is very important to understanding how missense mutations alter IF structure and function throughout all the IF classes.

2. Introduction, lines 47-52. For the description of lamin assembly, the authors cite their own recent work and that of Strelkov's group. I think you should additionally cite the field's most recent comprehensive review on the assembly mechanisms of intermediate filaments (Eldirany SA, et al. Recent insight into intermediate filament structure. *Curr Opin Cell Biol.* 2021 Feb;68:132-143) as an additional resource for those readers who wish to know more about the entire IF family.

3. Supplementary Figure 1 and Results, lines 82 – 85. The figure shows size-exclusion chromatography traces, but there is no validation in this figure of the text statement that S143F remained a dimer like the wt protein 1-300. The authors need to either show this fact through SEC-MALS data with molecular weights, or SEC data with molecular weight standards, or cite/describe the Ahn 2019 paper where they actually include SEC-MALS data.

4. Supplemental Figure 3B: this figure shows a superposition of WT and S143F 1-300 crystal

structures. The text, or figure legend, needs to specify methods of alignment as well as report root mean square deviation (rmsd) of the superposition.

5. The figure legend for Fig. 3 is incorrect, as it lists a-c panels and the statement for panel b is short, with no corresponding figure. Furthermore, there is no mention in the text (Results lines 104-113) about the authors' interpretation of the 50 mM NaCl vs 150 mM NaCl pulldown experiments. Why did the authors use two NaCl concentrations and what does this data tell them about the nature of the interaction?

6. Lines 126-135 and Figure 5: this section and figure would be improved with a total solvent accessible surface area for the binding interface, as well as a breakdown of the polar, hydrophobic, acidic, basic SASAs of the interface. Usually when talking about surfaces a SASA calculation is necessary to place this particular interface in proper context.

7. What was the rationale for choosing the human fibrosarcoma cell line, as opposed to another cell line for cell-based studies? I don't see this rationale stated anywhere, other than possibly "as observed in previous reports," but it might be nice to add this rationale, even in Materials and Methods.

8. Results, line 147: "show the same binding pattern" really ought to read "show the same oligomeric pattern".

9. I get what the authors are saying with "homophilic proximity effects are enhanced," but I am not sure this phrase reads well for those not well-versed in IFs.

10. This is ultimately a crystal structure paper, yet there is not one image of the electron density from this structure. A figure showing the quality of the electron density around the S143F site/interface is a must. Especially since the authors' structure is reported at 4.0 Å resolution. In Supplemental Figure 4 you report atomic interaction distances, but nowhere in the paper do the authors mention that a major limitation of this study is the 4 Å resolution, at which it can be very difficult to accurately place side chains. This limitation should be discussed, and electron density maps provided to show the quality of the structure and provide a validation for the authors' interpretation of the structure.

11. Discussion, line 172 – error in phrase "present every A11" needs fixing.

12. Discussion, line 198 – I don't like the phrase "by which strengthened but still weak interactions..." I feel this is understating the significance of the work. In Figure 4 the authors show an octameric assembly from the crystal lattice. The S143F interface isn't weak, it is strong enough to drive an octameric assembly in your crystal. Yes, multiple F143 residues come together to amplify the hydrophobic strength of the interface. I would re-phrase the sentence to better reflect the hydrophobicity-driven oligomerization of the lamin from a positive viewpoint.

Moreover, on this concept of hydrophobicity-driven oligomerization of IFs, the authors fail to cite or discuss how the lamin work related to the K1-S233L/K10 octameric structure seen by the Bunick group in their 2019 EMBO manuscript. That group had a similar finding in that enhanced surface-exposed hydrophobicity, in this case from leucine, drove an octameric oligomer in the crystal lattice. The authors here should recognize that the capability for surface-exposed hydrophobic residues in IFs to generate higher-order oligomers has been characterized before, and use this as a springboard to delve deeper in your discussion. It enhances the validity and significance of the authors' findings in my opinion.

13. Supplementary Table 1: The resolution of the data is 4Å, but the $I/\sigma(I)$ is 3.6 in the highest resolution shell. This cutoff value has historically been 2. Why did your data processing not extend to $I/\sigma(I) = 2$? More modern methods of data processing utilizes CC1/2 for data inclusion, yet no CC1/2 values are reported here for this data set. Why? Please include them. The Wilson B-factor seems artificially low at 9.4, especially for a 4Å structure? Do you have an explanation for that? What are the average overall B-factors for this structure?

14. Can the authors please provide the structure coordinates and associated MTZ file for review?

Reviewer #2 (Remarks to the Author):

This work extends the results and modelling published by the same group in Ahn et al (2019) Nature Communications. This previous paper reported on the crystal structure of the 1-300 fragment of lamin A, which had significant novelty at the time. The same paper presented additional biochemical data such as cross-linking. As a result, the authors have come up with a molecular model of 3.5nm lamin filament. There, a rather unexpected ~ 16nm large overlap of the N- and C-terminal ends of lamin dimer (and the corresponding extensive A22 overlap) was proposed.

Unfortunately this conclusion was not supported at all by the more recent cross-linking-centered work by the Schirmer group (Makarov et al (2019) Nature Communications, <https://doi.org/10.1038/s41467-019-11063-6>).

The main, and very major, problem of the new submission that the authors effectively keep discussing the model proposed in their 2019 paper even though it contradicts the more recent data.

The new paper reports on the crystal structure of the S143F mutant of the 1-300 fragment, which is identical to the WT structure (published by the same authors before) except for a new crystal contact made by the introduced phenylalanines. Per se it is not too surprising that phenylalanines cluster in the crystal structure. However, no proof is provided that such clustering is indeed biologically relevant.

Reviewer #1 (Remarks to the Author):

Comment 1.

Comment 1-1) I think this manuscript erroneously uses the term “Crossing-over.” When I think of “crossing-over” in the scientific sense, I think of homologous recombination of DNA, where there is intercalation of DNA strands and exchange of genetic material. There is no evidence in the figures or data provided here that lamins truly “cross-over” or intercalate their helices or exchange any type of macromolecular material. Here we provided our responses for reviewer’s comments as point-by-point from.

Response: We acknowledged that the term “ Crossing-over” might be confusing in the interpretation of the manuscript. We changed the term “crossing-over” to “X-shaped interaction” in the revised manuscript.

Comment 1-2) What there is evidence of is increased hydrophobicity at the molecular surface due to a missense mutation, which then drives aberrant higher-order multimerization of the lamin compared to wild-type lamin. This aberrant multimerization leads to altered nuclear lamina and deformed nuclei.

Response: In this revision, we added the experimental results to address the reviewer’s comment. We experimentally compared the surface hydrophobicities of the wild-type and the S143F mutant lamin 300 fragments using the hydrophobic interaction chromatography. The result showed a higher hydrophobicity on the surface of the S143F mutant fragment than that of the wild-type fragment (see **Fig. 5D in the revised manuscript**).

Comment 1-3) In thinking about this issue in the larger context of IF biology, and using the authors' Figure 7 diagrams as a baseline, what is really described here is an “X-shaped lamin assembly process” or “X-shaped lamin multimerization” or “X-shaped lamin oligomerization.” This X-shaped phenomenon has been seen in IFs before – the Coulombe group (K5/K14) and the Bunick group (K1/K10) demonstrated with X-ray crystal structures of 2b domains of heteromeric keratins that the type I keratins contain cysteines that lead to X-shaped oligomerization. The Coulombe group further showed this altered X-shaped K5/K14 has consequences for nuclear shape. I think the authors miss a big opportunity here to discuss more deeply the X-shaped structures in IF biology, their significance to the nucleus, and how their lamin structure is similar and dissimilar to the keratins (for example, the keratin and lamin data both have affect on the nucleus, which speaks to the importance of IFs in maintaining not just cell integrity, but nuclear integrity; they are different in that the keratin X-shapes come from disulfide bond formation, whereas the lamin is increased hydrophobic surface-exposed area from the aromatic mutant residue). Furthermore, the discussion of the disulfide data in the keratin work helps justify the authors decision to introduce S149C into their structural analysis. I encourage the authors to think about this, and re-phrase “crossing-over” to something that genuinely reflects the work to date in the IF field. This X-shaped aggregation is very important to understanding how missense mutations alter IF structure and function throughout all the IF classes.

Response: We appreciate the reviewer's comments, which significantly improved our manuscript. We mentioned the crystal structure of the keratin (K1/K10 and K5/K14), similar X-shaped interaction between heterodimers mediated by the disulfide bond. We discussed the X-shaped interactions between lamin S143F and keratins in terms of their similarities and differences. **(Discussion; page 10, line 188 – page 11, line 195)**

Comment 2. Introduction, lines 47-52. For the description of lamin assembly, the authors cite their own recent work and that of Strelkov's group. I think you should additionally cite the field's most recent comprehensive review on the assembly mechanisms of intermediate filaments (Eldirany SA, et al. Recent insight into intermediate filament structure. *Curr Opin Cell Biol.* 2021 Feb;68:132-143) as an additional resource for those readers who wish to know more about the entire IF family.

Response: We added the references for understanding the manuscripts in the revised manuscript, as suggested by the reviewer. We are happy to add the comprehensive review the reviewer mentioned in the revised manuscript. **(page 3, line 42)**

Comment 3. Supplementary Figure 1 and Results, lines 82 – 85. The figure shows size-exclusion chromatography traces, but there is no validation in this figure of the text statement that S143F remained a dimer like the wt protein 1-300. The authors need to either show this fact through SEC-MALS data with molecular weights, or SEC data with molecular weight standards, or cite/describe the Ahn 2019 paper where they actually include SEC-MALS data.

Response: To address the reviewer's comment, we added the SEC-MALS result, representing that the oligomeric state of lamin 1-300 WT and S143F is dimer. **(see Supplementary Fig. 1 in the revised manuscript)**

Comment 4. Supplemental Figure 3B: this figure shows a superposition of WT and S143F 1-300 crystal structures. The text, or figure legend, needs to specify methods of alignment as

well as report root mean square deviation (rmsd) of the superposition.

Response: We appreciate these helpful comments. We added the rmsd values of the superposition of the wild-type and S143F crystal structures in the revised manuscript. We added the detailed methods of structure alignment in the revised manuscript. (**page 6, line 90-92 and page 7, line 101-104; Supplementary Fig. 3 and 4, materials and methods**)

Comment 5. The figure legend for Fig. 3 is incorrect, as it lists a-c panels and the statement for panel b is short, with no corresponding figure. Furthermore, there is no mention in the text (Results lines 104-113) about the authors' interpretation of the 50 mM NaCl vs 150 mM NaCl pulldown experiments. Why did the authors use two NaCl concentrations and what does this data tell them about the nature of the interaction?

Response: Thank you for your kind comment. We corrected the mistake and we removed the statement for panel b. We observed a different binding affinity between lamin 300 fragment and the C-terminal fragment of coil 2 depending on the buffer conditions. The lamin 300 fragment is strongly bound to the C-terminal fragment of coil 2 at a relatively low NaCl concentration of 50 mM and shows weak binding affinity to the C-terminal fragment of coil 2 at a relatively high NaCl concentration of 150 mM. We added this description in the revised manuscript. (**page 7, line 112-116**)

Comment 6. Lines 126-135 and Figure 5: this section and figure would be improved with a total solvent accessible surface area for the binding interface, as well as a breakdown of the polar, hydrophobic, acidic, basic SASAs of the interface. Usually when talking about surfaces a SASA calculation is necessary to place this particular interface in proper context.

Response: We improved Figure 5 by adding the SASA analysis data. We confirmed that the hydrophobic surface area of the S143F mutant structure was increased by 1.5% compared to the wild-type. (page 8, line 138-140; Fig. 5C)

Comment 7. What was the rationale for choosing the human fibrosarcoma cell line, as opposed to another cell line for cell-based studies? I don't see this rationale stated anywhere, other than possibly "as observed in previous reports," but it might be nice to add this rationale, even in Materials and Methods.

Response: We believe that the HT1080 cell line data shows the nucleus morphology aging-related LMNA mutant because its nucleus is relatively large enough to clearly distinguish between the structures of nucleus and cytoplasm. We added the description in the main text. (page 9, line 149-151)

Comment 8. Results, line 147: "show the same binding pattern" really ought to read "show the same oligomeric pattern".

Response: We revised the sentence "We changed Ser149 to cysteine to mimic the disulfide-mediated X-shaped interaction since Ser143 and Ser149 are on the same interface for the X-shaped interaction (Supplementary Fig. 5)". (page 9, line 157-159)

Comment 9. I get what the authors are saying with "homophilic proximity effects are enhanced," but I am not sure this phrase reads well for those not well-versed in IFs.

Response: We re-phrase the sentence "it amplifies between adjacent lamin filaments." (page 10, line 167-169)

Comment 10. This is ultimately a crystal structure paper, yet there is not one image of the electron density from this structure. A figure showing the quality of the electron density around the S143F site/interface is a must. Especially since the authors' structure is reported at 4.0 Å resolution. In Supplemental Figure 4 you report atomic interaction distances, but nowhere in the paper do the authors mention that a major limitation of this study is the 4 Å resolution, at which it can be very difficult to accurately place side chains. This limitation should be discussed, and electron density maps provided to show the quality of the structure and provide a validation for the authors' interpretation of the structure.

Response: Thank you for your constructive comment. To address the reviewer's comment, we added a figure of the electron density map showing the S143F interface in Supplementary Fig 2. We recognized the limitations of the resolution of this structure, and an explanation related this limitation was added in the revised manuscript. We removed the atomic interaction distances in Supplementary Figure 4. (**page 9, line 154-157; Supplementary Fig. 2 and 5**)

Comment 11. Discussion, line 172 – error in phrase “present every A11” needs fixing.

Response: Thank you for your kind comment. We correct the mistake. (**page 11, line 198**)

Comment 12.

12-1) Discussion, line 198 – I don't like the phrase “by which strengthened but still weak interactions...” I feel this is understating the significance of the work. In Figure 4 the authors show an octameric assembly from the crystal lattice. The S143F interface isn't weak, it is

strong enough to drive an octameric assembly in your crystal. Yes, multiple F143 residues come together to amplify the hydrophobic strength of the interface. I would re-phrase the sentence to better reflect the hydrophobicity-driven oligomerization of the lamin from a positive viewpoint.

Response: We appreciate these helpful comments. We revised the sentence “In this study, we visualized the aberrant X-shaped interaction by S143F, which abolished the normal 3-D meshwork between the lamin filaments.” (page, 12, line 227-228)

12-2) Moreover, on this concept of hydrophobicity-driven oligomerization of IFs, the authors fail to cite or discuss how the lamin work related to the K1-S233L/K10 octameric structure seen by the Bunick group in their 2019 EMBO manuscript. That group had a similar finding in that enhanced surface-exposed hydrophobicity, in this case from leucine, drove an octameric oligomer in the crystal lattice. The authors here should recognize that the capability for surface-exposed hydrophobic residues in IFs to generate higher-order oligomers has been characterized before, and use this as a springboard to delve deeper in your discussion. It enhances the validity and significance of the authors’ findings in my opinion.

Response: We acknowledged that the introduction did not sufficiently provide the background and prospect of this research. We added the paragraph describing hydrophobicity-driven oligomerization of IFs, as suggested by the reviewer. (See **Introduction; page 4, line 55-61 and Discussion; page 11, line 211 – page 12, 216**).

Comment 13. Supplementary Table 1: The resolution of the data is 4Å, but the $I/\sigma(I)$ is 3.6 in the highest resolution shell. This cutoff value has historically been 2. Why did your data processing not extend to $I/\sigma(I) = 2$? More modern methods of data processing utilizes $CC1/2$

for data inclusion, yet no CC1/2 values are reported here for this data set. Why? Please include them. The Wilson B-factor seems artificially low at 9.4, especially for a 4A structure? Do you have an explanation for that? What are the average overall B-factors for this structure?

Response: Thank you for your constructive comment. We added CC 1/2 values in the revised manuscript. The resolution of the data was improved to 3.7 Å considering the I/sig(I) and CC1/2 values. However, the Wilson B-factor is still low at 11.07 and the overall B-factor is 37.38 (**Supplementary table. 1**). Since Wilson B-factor reflects the overall orderness of the crystal, we believe that the orderness of our crystals may be high. However, we believe that the high solvent contents (over 70%) seemed to weaken the overall diffraction quality.

Comment 14. Can the authors please provide the structure coordinates and associated MTZ file for review?

Response: We provide the structure coordinate and associated MTZ file.

Reviewer #2 (Remarks to the Author):

This work extends the results and modelling published by the same group in Ahn et al (2019) Nature Communications. This previous paper reported on the crystal structure of the 1-300 fragment of lamin A, which had significant novelty at the time. The same paper presented additional biochemical data such as cross-linking. As a result, the authors have come up with a molecular model of 3.5nm lamin filament. There, a rather unexpected ~ 16nm large overlap of the N- and C-terminal ends of lamin dimer (and the corresponding extensive A22 overlap) was proposed.

Comment 1. Unfortunately this conclusion was not supported at all by the more recent cross-linking-centered work by the Schirmer group (Makarov et al (2019) Nature Communications, <https://doi.org/10.1038/s41467-019-11063-6>).

The main, and very major, problem of the new submission that the authors effectively keep discussing the model proposed in their 2019 paper even though it contradicts the more recent data.

Response: We realized that we insufficiently explained the more recent cross-linking-centered work by the Schirmer group (Makarov et al (2019) Nature Communications, <https://doi.org/10.1038/s41467-019-11063-6>). In this revision, we significantly rephased our Introduction to add the descriptions of the works by the Schirmer group. Recently, we succeeded in determination of a crystal structure covering the coil 2 region fully representing the A22 interaction, which directly supports the assembly model

proposed by Schirmer group at the atomic resolution. In our next paper, we will provide the structural evidence supporting the results by Schirmer group at the atomic resolution with the clarification of our previous contradicting results. In this revision, we changed ‘eA22’ to original ‘A22’, and significantly rephrased the Introduction part. **(page 4, line 47-54)**

Comment 2. The new paper reports on the crystal structure of the S143F mutant of the 1-300 fragment, which is identical to the WT structure (published by the same authors before) except for a new crystal contact made by the introduced phenylalanines. Per se it is not too surprising that phenylalanines cluster in the crystal structure. However, no proof is provided that such clustering is indeed biologically relevant.

Response: The results of the increased hydrophobicity by S143F might not be surprising. However, it is surprising that the increased hydrophobicity affected the 3D-meshwork structures between the filament, not in the formation of the filament. We believe that our structural implication of the mutant structure can be expanded to a common assembly mechanism explaining keratin and other IFs, as the reviewer #1 recognized.

In this revision, we provided the experimental evidence representing the increased surface hydrophobicity. **(see Fig. 5D in the revised manuscript)**

REVIEWERS' COMMENTS:

Reviewer #1 (Remarks to the Author):

Ahn and colleagues have performed a substantial revision to their manuscript examining the crystal structure of lamin A/C S143F variant. I appreciate the authors' switch to X-shaped, which I think is the most accurate current description. Furthermore, steps were made to discuss prior keratin findings and enhance the hydrophobicity analysis. The manuscript in its current form does contain typos, and i encourage polishing of the english language. For example, page 9, lines 149-150 should read: "cells have been widely."

Reviewer #1 (Remarks to the Author):

Comment 1.

Ahn and colleagues have performed a substantial revision to their manuscript examining the crystal structure of lamin A/C S143F variant. I appreciate the authors' switch to X-shaped, which I think is the most accurate current description. Furthermore, steps were made to discuss prior keratin findings and enhance the hydrophobicity analysis. The manuscript in its current form does contain typos, and i encourage polishing of the english language. For example, page 9, lines 149-150 should read: "cells have been widely."

Response:

Thank you for your kind and helpful comments. We corrected all the typos and grammatical mistakes through the professional English Editing service. We attached the certificate of English proofreading.